# The Emerging Role of Venetoclax-Based Treatments in Acute Lymphoblastic Leukemia

**DOI:** 10.3390/ijms231810957

**Published:** 2022-09-19

**Authors:** Shlomzion Aumann, Adir Shaulov, Arnon Haran, Noa Gross Even-Zohar, Vladimir Vainstein, Boaz Nachmias

**Affiliations:** Department of Hematology, Hadassah Medical Center and Faculty of Medicine, Hebrew University of Jerusalem, Jerusalem 91120, Israel

**Keywords:** venetoclax, acute lymphocytic leukemia, BCL-2

## Abstract

Venetoclax, a B-cell lymphoma (BCL-2) inhibitor, in combination with hypomethylating agents has become the new standard of care in elderly and unfit patients with acute myeloid leukemia, with significantly improved overall survival and quality of life. Studies of venetoclax combined with high-dose chemotherapy are emerging with evidence of higher rates of molecular remission. Recently, a growing number of publications bring forth the use of venetoclax in patients with acute lymphoblastic leukemia (ALL). In the current review, we present the biological rationale of BCL-2 inhibition in ALL, how the interplay of BH3 proteins modulate the response and the current clinical experience with various combinations.

## 1. Introduction

The B-cell lymphoma (BCL-2) family of proteins are key regulators of the intrinsic cell death pathway. BCL-2-mediated apoptotic signaling features distinct anti-apoptotic proteins (BCL-2, BCL-X_L_, MCL-1) and pro-apoptotic proteins (BAX, BAK), along with BH3-only proteins, that further regulate these interactions. Following various stimuli, the activation of BH3-only proteins BAD, NOXA, BIM, and BID inhibits BCL-2, thus releasing the inhibition of pro-apoptotic effector molecules BAX and BAK, which leads to the formation of pores within the outer mitochondrial membrane, the release of cytochrome and caspase activation that mediates apoptosis of the cell [1].

Dysregulation of BCL-2 proteins has been demonstrated in many hematological malignancies, including the hallmark translocation t (14; 18) involving BCL-2 in follicular lymphoma, but also in other non-Hodgkin lymphomas, multiple myeloma, chronic lymphocytic leukemia (CLL) and acute myeloid leukemia (AML) [2,3].

This led to the development of BH3-mimetics that bind BCL-2 and release pro-apoptotic proteins such as BIM, leading to apoptosis induction (Figure 1a). ABT-737 (navitoclax) inhibits BCL-2 as well as BCL-X_L_ and showed preclinical efficacy; however, its clinical use has been limited due to thrombocytopenia, which is mainly related to the concomitant inhibition of BCL-X_L_ [4]. Venetoclax, a more selective BCL-2 inhibitor, has been widely incorporated in the treatment of CLL and AML, changing long-practiced treatment paradigms [5,6,7,8].

Similar to CLL and AML, BCL-2 and BCL-X_L_ overexpression was reported in acute lymphocytic leukemia (ALL), and in recent years, emerging preclinical and clinical reports have highlighted the potential efficacy of venetoclax and other BCL-2 family inhibitors in the treatment of ALL, including T-ALL, B-ALL, TCF3-HLF ALL and MLL-AF4 ALL [9,10,11]. Our growing understanding of the interplay of various BCL-2 family members and BH3-only proteins promotes novel combinations and better-tailored treatment for ALL patients. In this review, we will summarize the current understanding of the biological rationale of targeting specific BCL-2 family proteins in ALL and the current accumulating clinical data.

## 2. Dysregulation of BCL-2 Proteins in ALL

Overexpression of BCL-2 has been widely reported in ALL. Certain molecular and cytogenetic characteristics have been shown to upregulate BCL-2, suggesting these might be especially sensitive to BCL-2 inhibition. B-ALL with aberrations in mixed-lineage leukemia (MLL) genes, such as t(4; 11), are recognized as a high risk leukemia and often are resistant to conventional therapies, with dismal prognosis [12]. The resulting fusion gene MLL/AF4 was shown to directly bind to the BCL-2 gene and upregulate its expression, whereas other BCL-2 family members, such as MCL1, were not affected [13]. Indeed, t (4; 11) ALL cell lines and primary samples show a higher expression of BCL-2 [14,15]. MLL-rearranged B-ALL cell lines were sensitive to venetoclax in vitro, with resulting apoptosis induction, as compared to wild-type cell lines. Venetoclax was also able to decrease blast counts in the peripheral blood and tumor volume in a mouse xenograft model with these cell lines. Interestingly, disruption of MLL-fusion-driven BCL-2 expression has been proposed as a major mechanism of action for the bromodomain inhibitor I-BET151 [16].

Another well described translocation in ALL is the t (17; 19) translocation, which results in the fusion protein TCF3-HLF and constitutes a subtype of B-ALL with extremely poor prognosis. Among other targets, TCF3-HLF upregulates BCL-2 [17]. Treatment of TCF-HLF-positive B-ALL samples with venetoclax also resulted in the rapid reduction of tumor burden in a mouse xenograft model, linking the elevated expression of BCL-2 and sensitivity to its inhibition [10,18].

Hypodiploid B-ALL is also characterized by the elevated expression of BCL-2 and BIM and low levels of BCL-X_L_, and were highly sensitive to both navitoclax and venetoclax, suggesting that the balance of BCL-2/BCL-X_L_ expression influences the sensitivity to specific BCL-2 inhibition [19]. Similar results were observed in vivo in mouse xenograft models. Venetoclax treatment induced a rapid reduction in leukemia blasts in peripheral blood, which was durable in 80% of the xenografts. Near-haploid B-ALL showed a more significant response to venetoclax, with all treated mice showing undetectable peripheral blood blasts throughout the 60-day period of the trial. Notably, control Ph+ and Ph-like xenografts did not respond [19]. In accord, studies of Ph+ and Ph-like ALL showed a reduced sensitivity to BCL-2 inhibition compared with Ph- cells [20]. This might be partially explained by a higher BCL-W and lower BCL-2 expression. Endogenous MCL-1, BCL-2 and BCL-X_L_ are expressed in unmanipulated BCR-ABL-transformed B-ALL cells; however, only the loss of MCL-1 resulted in apoptosis induction. Moreover, loss of MCL1 delayed the onset of BCR-ABL-transformed leukemia in a mouse xenograft model, with a strong selection against clones with the MCL-1 deletion [21], suggesting MCL-1 may be a better target for inhibition in Ph+ ALL.

### Interplay of Other BH3 Proteins

The correlation between BCL-2 expression and sensitivity to BCL-2 inhibition is not straightforward. BH3 profiling allows us to better identify the dependence of the cell on a specific anti-apoptotic BCL-2 protein, with a possibly better correlation to drug sensitivity [10]. Initial studies with BH3 profiling predicted that ALL cells are dependent on BCL-2 and would be sensitive to its inhibition [22]; however, ALL cells display variability in their sensitivity to BCL-2 inhibition, suggesting other BCL-2 pro- and anti-apoptotic proteins (e.g., BCL-X_L_, MCL-1) as well as BH3-only proteins (BIM, BAD) modulate the response.

*MCL-1.* As pro-apoptotic molecules are released upon BCL-2 inhibition, other proteins such as MCL-1 potentially sequester them and prevent apoptosis. Accumulating evidence suggests that navitoclax-induced apoptosis requires the release of BIM and its binding and neutralization of MCL-1 [23]. Immunoprecipitation of BIM in cells treated with navitoclax showed a decrease in binding of BCL-2 and BCL-X_L_, and an increase in MCL1 interaction [21]. Low levels of MCL-1 were correlated with sensitivity to venetoclax and navitoclax [24], whereas the exogenous expression of MCL-1, including in primary ALL cells, reduced the sensitivity of the cells to navitoclax and rendered venetoclax ineffective [14]. Interestingly, treatment with venetoclax was shown to induce PI3K/AKT/mTOR activation and the upregulation of MCL1 and BCL-X_L_ levels as a possible mechanism of venetoclax resistance [25]. Thus, the balance of these proteins determines the sensitivity to apoptosis induction and can rationalize drug combinations to tip the balance toward apoptosis of the malignant cell.

*BCL-X_L_*. BCL-X_L_ has also been shown to correlate with sensitivity to BCL-2 inhibition, with high BCL-2/low BCL-X_L_ having the best response to venetoclax in vivo [26]. Navitoclax, which also inhibits BCL-X_L_, had a broader activity against the ALL xenograft than venetoclax, although there was a correlation between the response to the two drugs in individual xenografts, suggesting BCL-2 inhibition also contributes to the cytotoxic effect. Chonghaile et al. used BH-3 profiling in various T-ALL cell lines and primary cells and showed a dependence on BCL-X_L_, with a higher sensitivity to navitoclax that inhibits BCL-X_L_ and BCL-2 compared to the more selective venetoclax. Interestingly, early T-cell progenitor (ETP)-ALL was found to be more BCL-2 dependent. This finding is in correlation with the pattern of BCL-2/BCL-X_L_ expression alongside T-cell maturation. Early progenitor double-negative CD4/CD8 T-cells express abundant BCL-2 and little BCL-X_L_, whereas double-positive cells express higher levels of BCL-X_L_. In accordance, patient-derived xenografts with ETP T-ALL cells were sensitive to both venetoclax and navitoclax, whereas the typical T-ALL xenografts were more sensitive to navitoclax [27].

*BIM.* Venetoclax binds to BCL-2, releasing BIM which recruits BAX/BAK in active conformation to the mitochondrial membrane. BAX/BAK homo-oligomerization leads to mitochondrial outer membrane permeabilization, cytochrome C release and the induction of caspase-mediated apoptosis. BIM was also shown to neutralize MCL-1, reducing another anti-apoptotic barrier. Thus, low BIM levels might render BCL-2 inhibition ineffective. Indeed, lower levels of BIM in Ph+ and Ph-like leukemia cells correlate in vivo with resistance to venetoclax and the knockout of BIM-induced resistance to venetoclax [19].

## 3. Rationale for Drug Combinations with BCL-2 Inhibition

*Modulating BCL-2 family proteins*. As elaborated above, MCL-1, BCL-X_L_ and BIM levels can alter the response to BCL-2 inhibition. Studies of concomitant inhibition or modulation of protein levels demonstrate a synergistic effect (Figure 1b). Combining S63845, a selective inhibitor of MCL-1, with venetoclax had a strong synergistic effect in apoptosis induction in multiple T-ALL cell lines in vitro and in vivo [28]. Similarly, silmitasertib, a potent oral inhibitor of casein kinase 2 that reduces MCL-1 levels through enhanced proteosome degradation, showed a synergistic apoptosis induction combined with venetoclax [29]. Combined BCL-2 and MCL-1 inhibition also demonstrated a potent activity in Ph+ ALL xenografts, with a similar efficacy to dasatinib, a tyrosine kinase inhibitor. The combination was also effective in the Ph-like CRLF2-rearranged subtype of B-ALL, suggesting a similar biological phenotype [29,30]. Treatment of Ph+ ALL with the tyrosine kinase inhibitor (TKI) resulted in a decrease in MCL-1 expression, whereas BCL-2 and BCL- X_L_ were not affected. In accord, TKI combined with navitoclax or venetoclax showed a synergistic effect with increased apoptosis and reduced clonogenicity, whereas the depletion of BIM rendered the cells partially resistant to TKI, again demonstrating its possible role in neutralizing MCL-1 [21,31,32]. Dasatinib and ponatinib were also shown to induce the Lck/YES novel tyrosine kinase (LYN), which upregulates BIM and inhibits MCL-1, possibly contributing to the sensitivity to venetoclax [33] and further supporting TKI-BCL-2 inhibition combinations.

BET bromodomain inhibition also affects BIM through the induction of BIM expression and reduced BCL-2 expression, with a resulting synergistic effect from venetoclax and the BET bromodomain inhibitor JO1 in T-ALL cell lines and the patient-derived xenograft [34].

Finally, the co-inhibition of BCL-2 and BCL-X_L_ with a selective inhibitor (A-1155463), or by combining venetoclax and navitoclax, also resulted in the synergistic killing of ALL cells [26].

*Chemotherapy-based combinations*. BCL-2 silencing experiments, as well as results from preclinical xenograft models, suggest that BCL-2 promotes chemoresistance and cell survival; however, it does not act as a key driver oncogene, and thus, venetoclax might not be effective as monotherapy, but rather more effective in combination with other apoptosis inducers. Indeed, combining venetoclax and chemotherapy showed a prominent synergistic effect [13], resonating with the clinical data presented below. In addition, early reports have shown navitoclax can enhance the cytotoxicity in combination with L-asparaginase, vincristine and dexamethasone, which was observed in cell lines and xenograft models [35].

## 4. Clinical Experience with Venetoclax-Based Combinations

Based on the preclinical evidence of lymphoblastic cells’ dependence on BCL-2 and BCL-X_L_, several groups investigated the use of BH3-mimetics in ALL. A summary of significant clinical trials is presented in Table 1.

### 4.1. Venetoclax with Chemotherapy

The first report of clinical response to venetoclax in ALL was published by Numen et al. in 2018. The patient, a 71 yo female, was diagnosed with early T-cell precursor (ETP)-ALL; she received four treatment lines (hyper-CVAD, nelarabine, L (liposomal)-vincristine, and palbociclib with dexamethasone in a clinical trial) but was refractory. She was then started on mini-CVD (hyper-CVAD minus doxorubicin, with alternating methotrexate and cytarabine) in combination with venetoclax 400 mg. The patient achieved morphological complete remission (CR) after the second cycle with positive minimal residual disease (MRD). At the time of report, the patient was still in remission. The authors also presented a 75 yo patient with non-ETP T-ALL, who was treated with the same regimen and achieved morphologic CR with positive MRD, but his response was transient, with a full-blown relapse 3 weeks after the third cycle. They showed a strong BCL-2 expression in the first patient’s blasts, but not in the second patient’s blasts [36]. Based on this report and the preclinical data of ETP subtype susceptibility to BCL-2 inhibition, a phase I clinical trial testing the combination of venetoclax and mini-CVD in adults with Ph-neg B- and T- (including ETP) ALL was initiated at the Dana-Farber Cancer Institute and MD Anderson Cancer Center. The results were presented in the 2019 ASH annual meeting and included two cohorts: 10 previously untreated older (>60) patients, and 8 patients 18 and older with relapsed or refractory disease (r/r). Venetoclax was administered on days 1 to 21 of a 28-day cycle at two dose levels of 400 mg (3 patients) and 600 mg (15 patients), concurrent with mini-CVD, and followed by allogeneic stem cell transplantation (allo-SCT) or venetoclax + POMP (6-mercaptopurine, vincristine, methotrexate and prednisolone) maintenance. All the previously untreated patients responded to therapy, nine achieved CR with MRD negativity and one patient achieved a partial response (PR). Four patients remained on treatment whereas six patients had allo-SCT. With a median follow up of 11.3 months, none of the patients relapsed until the results were published. Among the eight patients with r/r disease, three achieved a CR, two of them had MRD negativity and two had ongoing responses for up to almost 4 months. In this work the most common grade 3 adverse event (AE) was febrile neutropenia (39%), and there were two 4 grade AEs, both infectious [37].

In the ASH 2021 annual meeting, a phase II study that examined the addition of venetoclax to mini-CVD reported the results of 23 patients with Ph-neg ALL, either > 60 years with a previously untreated disease, or > 18 years with r/r disease. In this study, venetoclax was given at a 400 mg dose on days 1–14 of cycle one, and on days 1–7 of the proceeding cycles. In addition, CD20+ B-ALL patients received rituximab, and nelarabine + PEG-asparaginase was incorporated into consolidation and maintenance for T-ALL patients. Patients were intended to proceed to allo-SCT or to receive VCR, prednisolone and venetoclax maintenance. Among the four previously untreated patients, three attained CR with MRD negativity and one achieved PR; all responses were achieved after one cycle. At a median follow up of 1 year, all CR patients remained in remission. Among the relapsed cohort, 11 (65%) responded (CR/CRi) and 3 achieved MRD negativity, and all responses were within the first two cycles. The reported median progression free survival (PFS) in the r/r cohort was 6 months, and overall survival (OS) was 7 months [38]. The conclusion of these prospective studies was that mini-CVD with venetoclax is a safe and effective regimen for Ph-neg ALL patients.

With the scarcity of alternatives, especially in the r/r setup, several reports on ETP or T-ALL patients salvaged with chemotherapy in combination with venetoclax were published. A case series of five adult patients with r/r ETP-ALL treated with venetoclax in combination with chemotherapy reported an 80% (4/5) CR rate. The patients were treated for 2–4 cycles with venetoclax and hyper-CVAD (2 pts), mini-CVD (2 pts) or nelarabine (1 pt). MRD negativity was achieved in two cases, where both patients were referred to allo-SCT. One of the CR-MRD+ patients was referred to allo-SCT and was in remission at a 3 mo post-transplant follow up. The other CR-MRD+ patient was not eligible for SCT and received venetoclax with vincristine and prednisone maintenance and remained in remission for 5 months until relapse. None of the patients in this series developed tumor lysis syndrome under venetoclax treatment, and there were no reported toxicities other than transient cytopenia [39]. In another report from China, a 25 yo male with relapsed ETP-ALL was successfully treated with venetoclax 400 mg and HAG chemotherapy (aclacinomycin, cytarabine and G-CSF). CR was attained after one cycle and the patient went to allo-SCT, and on a follow up 5 months post-transplant was in remission [40].

Two retrospective analyses from MD Anderson described their experience with venetoclax combinations in ALL. In a report of 13 adult patients (median age 46) treated with venetoclax and chemotherapy for r/r T-ALL, 60% of evaluable patients achieved CR. The concurrent chemotherapy regimens included hyper-CVAD, asparaginase with vincristine or nelarabine, decitabine or FLAG-Ida. Venetoclax was given in a 100–400 mg dose, for a median of 21 days per cycle. The main AE observed was prolonged cytopenia, there were no deaths within 30 days of venetoclax administration and all subsequent deaths were due to disease progression. Among the eight responders, the median PFS was 4 months, and the median OS of the entire cohort was 7.7 months [41]. In an analysis of 18 pediatric and young adults (<22) with a median age of 20 and who received venetoclax in combination with chemotherapy, 11 (61%) responded (CR/Cri) and all achieved MRD negativity. Most of the patients received hyper-CVAD or CVD as concurrent chemotherapy, and 400 mg was the venetoclax dosage for the majority of patients. Among the 13 T-ALL (including ETP-ALL) patients, the response rate was 77%. The most common AE was thrombocytopenia, with 22% requiring a dose adjustment. Neutropenia and febrile neutropenia were 50% and 22%, respectively. With a median follow up of 12.11 mo, the OS is 9.14 mo, and PFS is 7.3 mo [42].

Few other chemotherapeutic regimens were examined in combination with venetoclax in prospective studies. In a phase 1 dose escalation study of venetoclax with L-vincristine, 18 adult patients (median age 42) with r/r B or T-ALL were included. Patients received 2 weeks of a lead-in phase of venetoclax in increasing dosage until reaching 400–800 mg daily. L-Vincristine was added to the treatment when the venetoclax full dose was reached. Grade 3 or more AEs were reported in 89% of the patients, with cytopenia being the most common. One patient experienced tumor lysis syndrome. Two patients had rapidly progressing disease under the treatment. Two patients had to stop the treatment due to toxicity. Four patients (22%) attained CR under the treatment, and two of them had MRD negativity [43].

The largest study published to date is a phase 1, multicenter study that examined the combination of venetoclax and navitoclax with chemotherapy (VCR, dexamethasone +/- Peg-asparaginase). In this study, 47 patients, 12 of whom were under 18 years of age, with r/r ALL were included. Patients received venetoclax 400 mg daily with escalating doses of navitoclax (25–100 mg daily) in combination with chemotherapy. Thirty-seven patients (78%) experienced serious AEs; the most common grade 3/4 AEs were febrile neutropenia (46.8%), thrombocytopenia (38.3%) and neutropenia (19%). Venetoclax and navitoclax were discontinued due to AEs in 11 patients, where 6 were considered related to venetoclax and 5 to navitoclax. During the study, 30 patients (63%) died, 26 of them from disease progression. A response (CR, CRi, CRp) was achieved by 28 patients (60%), including 9 children. The overall response rate (ORR) among those 18 yo and younger was 75%. MRD negativity was achieved by 16 patients (34%), where 6 of them were 18 and younger. A post hoc analysis of patients’ subgroups revealed no differences between B- or T-cell disease. The median duration of response was 4.2 mo, and OS was 7.8 mo. Correlative BH3 profiling demonstrated frequent BCL-2 and BCL-X_L_ dependency in ALL cells, and the capacity for dependency switching during treatment as a potential mechanism of resistance to BCL-2 inhibition [44].

The combination of chemotherapy and targeted therapies with venetoclax was also suggested in a recent report. A 26 yo r/r pro-BALL patient was treated with venetoclax (100 mg, days 3–7) and the anti-CD38 antibody daratumumab (days 2 and 10) with cyclophosphamide (150 mg/m^2^, days 1–3) and clofarabine (20 mg/m^2^, days 1–5). After prolonged agranulocytosis of 43 days, hematopoietic reconstitution with CR was attained. The patient remained MRD-negative for 5 months as he was waiting for allo-SCT. In a follow up 12 months after SCT, he was still in remission under venetoclax maintenance [45].

Taken together, the administration of venetoclax in combination with low-dose chemotherapy regimens for Ph-neg B or T-ALL seems to be feasible and effective. It is especially important in the setup of r/r ETP-ALL with the scarceness of available therapeutic options.

### 4.2. Chemotherapy-Free Regimens

Elderly patients with Ph-neg ALL have a dismal prognosis, with high rates of early death. A first report of chemotherapy-free regimens for the treatment of Ph-neg ALL included three newly diagnosed elderly patients that were ineligible for intensive chemotherapy. The patients were treated with venetoclax 100 mg on day 1, and 200 mg days 2–28, with prednisone 1 mg/kg days 1–21, and 0.5 mg/kg days 22–28. Two patients achieved CR after the first cycle, one of which had MRD negativity. No AEs were noted, and the clinical status of both patients was significantly improved under treatment. Nevertheless, both patients relapsed after 2 months, and died of active disease. The third patient was refractory to the venetoclax + prednisone regimen [46]. With the limitation of only three patients, this report warrants further investigations of the venetoclax + prednisone regimen for selective populations.

Venetoclax with azacitidine, which became the standard-of-care regimen for AML patients who are ineligible for intensive chemotherapy, has significantly changed the survival rates of the disease. In a thorough review of the literature, we have found four case reports of r/r T-ALL patients treated with the combination of venetoclax and decitabine, and one case series of five r/r T-ALL patients treated with venetoclax with azacitidine.

The first report from UCSF was a 20 yo male, initially diagnosed with AML, but in retrospect, had T-ALL. He achieved CR after induction chemotherapy and was consolidated with SCT, but relapsed 13 months after with a disease that was found to be T-ALL. He achieved CR with MRD negativity following treatment with venetoclax (800 mg, days 1–28) and decitabine (20 mg/m^2^, days 1–5), and continued the treatment for five cycles until his second allo-SCT [47]. The second report was of a 36 yo female with r/r ETP-ALL; she was given venetoclax 600 mg daily with similar doses of decitabine. Although she attained CR with MRD negativity after one cycle, the second cycle was interrupted due to cytopenia and systemic infections, which she eventually died of [48]. Two case reports of young women treated with venetoclax and decitabine for r/r ETP-ALL described CR after up to two cycles, with no severe AEs, and sustained remission 3 months after allo-SCT [49,50]. Venetoclax with azacitidine was given to five primary refractory T-ALL patients in China. Four patients achieved CR and remained in remission under treatment for 1–7 months at the time of report. The fifth patient achieved Cri, was referred to allo-SCT and died of a-GVHD. Cytopenia occurred in all five patients under the Ven-Aza treatment, with a median of 17 days to neutrophil recovery after initiation of therapy. No other grade 3/4 AEs were reported [51]. Preliminary data suggest that the synergy between venetoclax and HMA therapy may also be beneficial in ALL and can present an effective alternative for chemo-resistant patients, especially in elderly or frail patients.

Ph(+) ALL outcomes have greatly improved in the TKIs era, and the incorporation of blinatumomab in the treatments made chemotherapy-free regimens available [52]. Nevertheless, r/r Ph(+) ALL still carries a poor prognosis and improved approaches are needed. The chemotherapy-free combination of venetoclax with ponatinib and dexamethasone (VPD) for r/r Ph+ ALL was examined in a phase 1 study. Nine adult patients (median age 37) with at least one prior Tki treatment were included in the analysis. Two doses of venetoclax, 400 mg (three patients) and 800 mg (three patients), were investigated. The combination was given on a 28-day cycle protocol. For CD20+ patients, rituximab was added. Most AEs were grade 1–2 and related to cytopenia. No dose-limiting toxicities were observed, and the maximum-tolerated dose of venetoclax was not reached. Five patients (56%) responded (CR/Cri), and all were in the 800 mg dose level. Four patients (44%) achieved complete molecular response (CMR). With a median follow up of 13.2 mo, two patients died from progressive disease, none have relapsed, and the median OS was not reached. All responding patients remained in response without allo-SCT [53]. In a retrospective analysis of 19 r/r Ph(+) ALL patients, the outcomes of a treatment with VPD were reported. After one cycle of treatment, 17 patients (89.5%) attained CR/Cri, with 14 patients having MRD negativity, and 8 patients had CMR. Grade 3 and more AEs were mostly due to myelosuppression with 73% neutropenia and 52% thrombocytopenia. Among 11 patients who continued with VPD, 7 subsequently relapsed, whereas among the patients who went to allo-SCT, only 1 of 6 patients relapsed. This suggests that bridging to allo-SCT in remission is warranted [54].

Finally, numerous ongoing trials are incorporating venetoclax and other BCL-2 family inhibitors in various regimens, including chemotherapy, immunotherapy (blinatumomab), hypomethylating agents and others (Table 2).

**Table 1 ijms-23-10957-t001:** Significant clinical studies with venetoclax-based combinations.

First Author, Year	Cohort	Design, Regimen	ORR	OS	Grade III/IV AE
Jain 2019 [37]	*n* = 18 Ph-neg ALL > 18 yo r/r or > 60 yo ND	Phase 1, Ven 400/600 mg with mini-hyper-CVD	r/r: 37.5% ND: 100%		39% FN, 17% hyperglycemia, 11% hypocalcemia, 1 pneumonia, 1 sepsis
Carpentier 2019 [41]	*n* = 13 r/r T-ALL adults	Retrospective, Ven 100–400 mg with chemotherapy/HMA	60%	7.7 mo	Prolonged cytopenia (percentage not reported)
Venugopal 2021 [38]	*n* = 23 Ph-neg ALL > 18 yo r/r or > 60 yo ND	Phase 2, Ven 400 mg with mini-hyper-CVD	65%	7.1 mo	Cytopenia (percentage not reported), 1 sepsis
Gibson 2021 [42]	*n* = 18 ALL age < 22	Retrospective, Ven 400 mg with chemotherapy	61%		89% thrombocytopenia, 53% neutropenia, 22% hyperbilirubinemia, 28% sepsis, 28% FN, 1 pneumonia, 1 coagulopathy, 1 mucosal infection
Pullarkat 2021 [44]	*n* = 47 r/r ALL, all ages	Phase 1, Ven + navitoclax with chemotherapy	59.6%	7.8 mo	46.8% FN, 38.3% neutropenia, 25.5% thrombocytopenia
Palmisiano 2021 [43]	*n* = 18 r/r Ph-neg ALL adults	Phase 1, Ven + L-vincristine	44%		67% neutropenia, 56% leukopenia, 50% anemia, 1 TLS
Short 2021 [53]	*n* = 9 r/r Ph(+) ALL adults	Phase 1, Ven + ponatinib + dex	56%		44% cytopenia
Wang 2022 [54]	*n* = 19 r/r Ph(+) ALL adults	Retrospective, Ven + ponatinib + dex	89.5%	400 days	73% neutropenia, 36% anemia, 52% thrombocytopenia

*n*—number of patients; Ph-neg—Philadelphia chromosome-negative; ALL—acute lymphoblastic leukemia; yo—years old; r/r—relapsed/refractory; ND—newly diagnosed; Ven—venetoclax; CVD—cyclophosphamide, vincristine and dexamethasone; HMA—hypomethylating agent; dex—dexamethasone; ORR—overall response rate; OS—overall survival; mo—months; AE—adverse event; FN—febrile neutropenia.

**Table 2 ijms-23-10957-t002:** Ongoing registered clinical trials with venetoclax-based treatments in ALL.

	Population	Intervention	Study Type	Primary Outcome	Primary Investigator
NCT05433532	Adults with newly diagnosed Ph+ ALL, AML and CML-AP/BP patients	Venetoclax, azacytidine, and flumatinib induction and consolidation	Phase 2, open-label	Complete molecular remission at end of cycle 2	Xiaowen Tang, The First Affiliated Hospital of Soochow University
NCT03826992	Children and young adults with R/R AML, MPAL, AUL, KMT2A-rearranged ALL, T-cell ALL or ETP-ALL	Venetoclax with CPX-351	Phase 1, open-label,	Feasibility, toxicity	John Perentesis, Children’s Hospital Medical Center, Cincinnati
NCT04029688	Children and young adults with newly diagnosed neuroblastoma, AML or ALL	Idasanutlin in combination with chemotherapy or venetoclax	Phase ½, open-label	Feasibility, toxicity	Hoffmann-La Roche
NCT04000698	Children with CD38+, CD184+ and Bcl2+, R/R AML or ALL	Haploidentical HSCT with conditioning chemotherapy including venetoclax	Phase 1/2, open-label	Engraftment at day + 30 after HSCT, ORR, PRR, toxicity, transplant-related mortality	Michael Maschan, Federal Research Institute of Pediatric Hematology, Oncology and Immunology, Moscow
NCT05292664	Children and young adults with R/R ALL and other hematological malignancies	Venetoclax combined with chemotherapy	Phase 1, open-label	Tolerated dose, toxicity	Andrew E Place, Dana-Farber Cancer Institute
NCT05386576	Adults with newly diagnosed ALL	Venetoclax combined with asparaginase-containing pediatric-inspired chemotherapy	Phase 1, open-label	Dose-limiting toxicity	Jae Park, Memorial Sloan Kettering Cancer Center
NCT05192889	Children with R/R ALL	Phase 1: chemotherapy with venetoclax and navitoclax Phase 2: venetoclax with either blinatumomab (for CD19+) or high-dose cytarabine and navitoclax (for CD19-)	Phase 1/2, open-label	MRD negative CR/Cri rate following induction, recommended phase 2 dose	Seth E. Karol, St. Jude Children’s Research Hospital
NCT05005299	Adults with hematological malignancies who are planned for ASCT	Venetoclax therapy prior to non-myeloablative conditioning with fludarabine and cyclophosphamide	Phase 1, open-label	Toxicity	David Ritchie, Melbourne Health
NCT05149378	Adults and young adults with R/R T-ALL	Venetoclax with azacitidine	Phase 2, open-label	ORR, CR rate	Sheng-Li Xue, The First Affiliated Hospital of Soochow University
NCT03181126(completed)	Adults and children with R/R ALL	Venetoclax with navitoclax and chemotherapy	Phase 1, open-label	Pharmacokinetics, toxicity	Abbvie
NCT05268003	Adults and children with R/R T-ALL	Ponatinib With mini-hyper-CVD and venetoclax	Phase 2, open-label	CR, Cri rate	Jain Nitin, MD Anderson Cancer Center
NCT05157971	Adults (18–54) with newly diagnosed B-ALL	Venetoclax with the C10403 regimen for induction and consolidation	Phase 1, open-label	Safety, maximal tolerated dose	Ibrahim T Aldoss, City of Hope Medical Center
NCT00501826	Children and adults with newly diagnosed T-ALL	Hyper-CVAD in combination with nelarabine, venetoclax and PEG-asparaginase	Phase 2, open-label	CRR, duration of remission, PFS, OS	Farhad Ravandi-Kashani, MD Anderson Cancer Center
NCT03319901	Newly diagnosed ALL >60 yo or R/R ALL >18 yo	Venetoclax with standard chemotherapy	Phase 1, open-label	Tolerated dose	Daniel DeAngelo, Dana-Farber Cancer Institute
NCT05182385	Adults with R/R Ph- B-cell precursor ALL	Venetoclax with blinatumomab	Phase 1/2, open-label	Tolerated dose, rate of complete molecular remission after 1 cycle	Nicola Goekbuget, GMALL study-group
NCT03576547	Adults with Ph+ or BCR-ABL+ R/R ALL or CML	Venetoclax, ponatinib and dexamethasone	Phase 1/2, open-label	Tolerated dose, ORR	Farhad Ravandi-Kashani, MD Anderson Cancer Center
NCT04752163	Adults with R/R AML or ALL	DS-1594b with or without zacytidine, venetoclax, or mini-HCVD	Phase 1/2, open-label	Tolerated dose, toxicity, CR/Cri rate,	Naval G Daver, MD Anderson Cancer Center
NCT05016947	Adults with R/R CD22+ B-ALL	Venetoclax with inotuzumab	Phase 1, open-label	Tolerated dose, toxicity	Marlise R Luskin, Dana-Farber Cancer Institute
NCT03504644	Adults with R/R ALL	Venetoclax with liposomal vincristine	Phase 1b/2, open -label	Tolerated dose, toxicity, CR/Cri rate	Neil Palmisiano, ECOG-ACRIN Cancer Research Group
NCT04872790	Adults with newly diagnosed or relapsed Ph+ ALL	Venetoclax with dasatinib, prednisone and rituximab	Phase 1b, open-label	Tolerated dose, toxicity	Jessica T Leonard, OHSU Knight Cancer Institute
NCT05376111	Adults and young adults with newly diagnosed T-ALL	Venetoclax with azacitidine	Phase 2, open-label	ORR	Sheng-Li Xue, The First Affiliated Hospital of Soochow University
NCT03808610	Adults with R/R ALL	Venetoclax with low-intensity chemotherapy (cyclophosphamide, cytarabine, methotrexate, PEG-asparaginase, vincristine, rituximab)	Phase 1/2, open-label	Tolerated dose, toxicity	Elias Jabbour, MD Anderson Cancer Center
NCT05054465	Adults with high-risk T-ALL post-remission	Venetoclax and navitoclax pre- and post-ASCT maintenance	Phase 1b/2, open-label	EFS	Ofir Wolach, Israeli Medical Association

Ph—Philadelphia chromosome, ALL—acute lymphoblastic leukemia, AML—acute myeloid leukemia, CML—chronic myeloid leukemia, AP—accelerated phase, BP—blast phase, MPAL—mixed-phenotype acute leukemia, AUL—acute undifferentiated leukemia, ETP—early thymocyte precursor, HSCT—hematopoietic stem cell transplantation, ORR—overall response rate, PRR—partial response rate, MRD—minimal residual disease, CR—complete remission, Cri—complete remission with incomplete count recovery, PFS—progression free survival, OS—overall survival, R/R—relapsed/refractory, EFS—event-free survival.

## 5. Discussion

Preclinical studies demonstrate that BCL-2 family proteins play a significant role in ALL blasts, including at early leukemia initiation stages. As data accumulate, two inherent questions arise: can we identify which ALL sub-types would respond to BCL-2 inhibition, and what drug combinations are optimal? Certain molecular subtypes such as MLL-rearranged, TCF3-HLF and hypodiploid are correlated with BCL-2 overexpression, with preclinical evidence supporting the use of venetoclax in these subtypes. Ongoing clinical trials were designed to specifically target these subtypes. Similarly, early T-cell ALL is more dependent on BCL-2 and shows sensitivity to venetoclax, including in small clinical series [27,36,39,40]. Ph+ ALL and Ph-like tend to be more dependent on other BCL-2 proteins such as MCL-1 and BCL-X_L_, with an expected lower response to venetoclax as a single agent [19]. MCL-1 and BCL-X_L_ can sequester the pro-apoptotic effectors that are released upon the binding of venetoclax to BCL-2, rendering it ineffective. Lessons from CLL and AML offer additional novel insights on mechanisms of resistance. p53 mutations in AML were shown to reduce BCL-2 levels and increase MCL-1 levels, promoting venetoclax resistance [55,56]. Additionally, BCL-2 levels’ independent mechanisms have also been described, such as the change in the upregulation of fatty acid oxidation [57]. Venetoclax in itself was shown to upregulate MCL-1 levels [25]. Our growing understanding of BCL-2 proteins’ interplay and experience in AML has prompted clinical trials adding venetoclax to commonly used chemotherapy protocols.

Preliminary studies of venetoclax added to mini-hyper-CVD showed encouraging results, including in the r/r setting [36]. Others have attempted to combine different agents with the aim to overcome the resistance mechanisms described above. Navitoclax, which inhibits both BCL-2 and BCL-X_L_, was used in combination with venetoclax in r/r ALL with a remarkable complete remission rate of 60% [44]. Adding venetoclax to hypomethylating agents was tried in r/r T-ALL, including p53 mutation [48,50]. The MCL-1 specific inhibitor also showed a synergistic effect in vitro with venetoclax and holds promise as a future chemotherapy-free combination. Similarly, TKI treatments in Ph+ ALL were shown to reduce MCL-1 levels, sensitizing the cell to venetoclax. Clinical data in Ph+ ALL patients, including in those with T315I mutations, is limited but encouraging [54]. As described in Table 2, there are numerous clinical trials studying the use of venetoclax in specific subtypes and combinations. The results of these trials will help us to better tailor BCL-2 inhibition to the specific patient.

## 6. Conclusions

Clinical experience of venetoclax in ALL is still emerging, but the current available data support the hypothesis of BH3-mimetic activity in ALL. Venetoclax-containing regimens are relatively safe and can achieve deep responses, although these are frequently transient, and therefore, should be utilized as a bridge to all-SCT in eligible patients. Our growing understanding of the interplay of BCL-2 family proteins in ALL will allow us to better tailor BCL-2 inhibitors and combinations to a specific subtype. Ongoing clinical trials will shape how we incorporate these agents in the coming years.

## Figures and Tables

**Figure 1 ijms-23-10957-f001:**
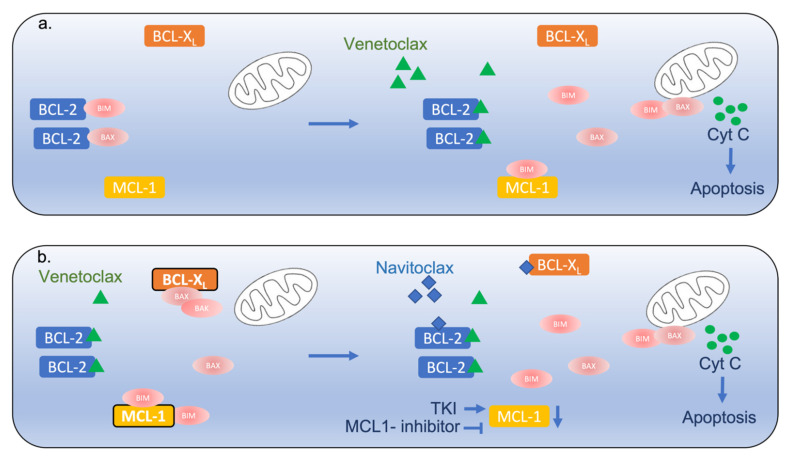
Interplay of BCL-2 family members in ALL. (**a**) Venetoclax binds to BCL-2 and displaces Bim and other BH3-only proteins (BAX), promoting activation of Bax and Bak and leading to their oligomerization and mitochondrial outer membrane permeabilization, cytochrome C release and caspase activation. (**b**) The upregulation of BCL-X_L_ and MCL-1 results in sequestration of these effectors and venetoclax resistance. This can be potentially overcome by co-targeting these molecules as presented. TKI—tyrosine kinase inhibitor, Cyt C—cytochrome C.

## Data Availability

Not applicable.

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
