# Peer review of "The Emerging Role of Venetoclax-Based Treatments in Acute Lymphoblastic Leukemia"

_ijms, 2022, doi:10.3390/ijms231810957_

Round 1

Reviewer 1 Report

Aumann et al in the present article clearly review the potential clinical activity of venetoclax in ALL. 

Few minor comments: 

·        Usually, venetoclax is written without capital letter because is not the brand name. This should also be applied to all other drug compounds (eg. Azacytidine and others)

· Line 32: Please add citations such as https://pubmed.ncbi.nlm.nih.gov/29561706/ and https://pubmed.ncbi.nlm.nih.gov/34320168/

·             In line 289, if correct, please read 0.5mg/kg instead of 0/5mg/kg.

· Line 323 please cite this important trial: https://pubmed.ncbi.nlm.nih.gov/33085860/

Author Response

We thank the reviewer for his positive remarks.

All requested corrections have been made including the added refrencses 

Reviewer 2 Report

This work is a nice and comprehensive overview of so far scarce experience and knowledge on Venetoclax testing in ALL. It is well organized, in a logical manner and readable. At present, I guess there is not anything else important that would be missing and should be added. Perhaps the authors could include a note on BH3 profiling, whether this has been tested in ALL and speculate whether this might be the way in ALL to predict the dependency on a particular BCL2 family protein and thereby vulnerability to the specific inhibitor.

The text should be checked for typing errors and missing words. There are few sentences that do not make sense as there seem to be something missing.

Author Response

We thank the reviewer for his positive remarks

We discuss BH3 profiling findings in several key papers. Please note Lines 87, 112,  266

The text underwent proof reading